# Rhino-Orbital-Cerebral Mucormycosis Complicated by Vision Loss in a Patient with Uncontrolled Diabetes: A Case Report

**DOI:** 10.3390/microorganisms13122695

**Published:** 2025-11-26

**Authors:** Martyna Lara, Patryk Hartwich, Anna Sepioło, Magdalena Namysł, Monika Bociąga-Jasik

**Affiliations:** 1Department of Infectious Diseases, University Hospital in Krakow, Department of Infectious and Tropical Diseases, Jagiellonian University Medical College, Jakubowskiego Str. 2, 30-688 Kraków, Poland; monika.bociaga-jasik@uj.edu.pl; 2Department of Otolaryngology, University Hospital in Kraków, Department of Otolaryngology, Jagiellonian University Medical College, Jakubowskiego Str. 2, 30-688 Kraków, Poland; phartwich@gmail.com; 3Department of Microbiology, University Hospital in Krakow, Jakubowskiego Str. 2, 30-688 Kraków, Poland; asepiolo@su.krakow.pl (A.S.); mnamysl@su.krakow.pl (M.N.)

**Keywords:** mucormycosis, zygomycosis, Mucorales, rhino-orbital mucormycosis, opportunistic infection, invasive fungal infection, diabetes mellitus, antifungals, surgical debridement

## Abstract

We present a case report of invasive fungal infection in an immunocompromised host, which required a multidisciplinary approach. Mucormycosis is a mold infection caused by a fungi belonging to the order Mucorales. Various forms of the disease have been described, and rhino-orbital-cerebral infection is the most common manifestation. Diabetes, corticosteroid use, malignancy, and a recent history of COVID-19 are well-established immunosuppressive factors that predispose individuals to mucormycosis. Our patient was a forty-five-year-old man with chronic pancreatitis and untreated diabetes mellitus. He presented with sinusitis extending into the right orbit and complicated by central retinal artery occlusion. On admission, the patient complained of three weeks of right-sided headache and eye pain followed by sudden vision loss. He was in good general condition, was alert, oriented, and afebrile. Endoscopic examination revealed the nasal cavity completely filled with pathological tissue displaying fungal morphology. Computed tomography and magnetic resonance imaging revealed a massive orbit infiltration with extraocular muscles and optic nerve invasion. The patient underwent urgent endoscopic debridement. Histopathological examination of the specimens confirmed fungal infiltration. Significant growth of *Rhizopus arrhizus* was obtained from tissue samples. The surgical procedure was followed by a prolonged antifungal therapy with intensive diabetes management.

## 1. Introduction

Mucormycosis is a life-threatening mold infection caused by a fungi belonging to the Mucorales order. Genera *Rhizopus, Mucor* and *Cunninghamella* are responsible for most of the cases. In susceptible hosts, inhaled spores typically give rise to rhino-orbital-cerebral or pulmonary disease, whereas direct percutaneous inoculation is associated with the cutaneous form. Mucorales fungi tend to invade the adjacent tissue and cause widespread destruction. Similarly to other systemic fungal infections, their propensity to enter blood vessels and cause local occlusion has also been widely documented [1].

Diabetes (particularly when complicated by ketoacidosis), corticosteroid use, malignancy, and a recent history of COVID-19 are well-established immunosuppressive factors that predispose individuals to mucormycosis. In such cases, host susceptibility is mediated by hyperglycemia, low pH, and endothelial damage [2]. In patients with no known immunosuppression, history of trauma or burns is a relatively common predisposing factor [3]. The pooled mortality of mucormycosis is estimated at 40–80% [3,4], with diabetes mellitus reported as an underlying condition in 46–68% of fatal cases [5,6].

Mucormycosis remains a diagnostic and therapeutic challenge. It used to be regarded as a rare disease, but, according to the data from a French surveillance program from the years 2012–2022, its epidemiology may be changing. The authors reported a rising incidence of hematological malignancy (attributable to better prognosis and more aggressive therapies) and a lower rate of diabetes in a cohort of mucormycosis patients. Additionally, incorporation of more sensitive molecular diagnostic methods may have contributed to increased prevalence of this fungal disease [7]. The largest global systematic reviews of the epidemiology of mucormycosis are based on analyses of case reports. The authors found that the prevalence of underlying conditions varies by region, with diabetes mellitus being a more common predisposing factor in Asia and the Middle East (reported in up to 75% of cases) than in Europe (10–37%), where hematological malignancies predominate. India, in particular, reported the highest number of COVID-19-related mucormycosis cases [3]. The Polish literature on the local prevalence of mucormycosis is limited. One report from 2020 focuses on patients with hematological malignancies and analyzes morbidity and associated factors [8].

Rhino-orbital-cerebral infection is the most common manifestation of mucormycosis. Patients usually present with symptoms of acute sinusitis with a fever accompanied by periorbital edema and visual disturbances [9]. On endoscopic examination, a typical black eschar (necrosis) on nasal mucosa or the palate can be seen in 40% of patients [10]. Treatment typically requires a combination of antifungals (amphotericin B and broad-spectrum azoles), surgical management, and reversal of immune suppression, whenever possible. Early detection of mucormycosis is crucial, as any delay in medical or surgical treatment negatively affects the prognosis [3,9].

## 2. Case Report

A forty-five-year-old man was admitted to the Otolaryngology Department of the University Hospital in Cracow, Poland, after being transferred from a secondary hospital because of severe sinusitis that had spread to the right orbit and required surgical management.

The patient complained of right-sided headache and eye pain followed by sudden vision loss in his right eye three weeks prior to admission. Moreover, he had lost 15 kg over the past 6 months. His past medical history was remarkable for untreated diabetes mellitus and a history of pancreatectomy due to acute pancreatitis 25 years before. He had never suffered from sinusitis before. He had not been taking any medication on a daily basis. He had no relevant travel history and denied recent trauma. He had been smoking cigarettes (10 pack-years) and drinking approximately 5 beers daily for at least 25 years. He was engaged in physical work, in construction.

At the patient’s initial presentation at the Otolaryngology Department of the secondary hospital, the magnetic resonance imaging (MRI) of orbits revealed infiltration of the right frontal, ethmoid and maxillary sinuses, extension into the right orbit’s soft tissue and extraocular muscles with optic nerve invasion (Figure 1). The ophthalmological examination revealed central retinal artery occlusion. The treatment with ceftriaxone, vancomycin, fluconazole, mannitol and methylprednisolone was initiated. Due to clinical suspicion of mucormycosis, the patient was transferred to the University Hospital in Cracow.

### 2.1. Endoscopic and Clinical Examination

On admission, the patient was in a good general condition, he was alert, oriented, afebrile. On ophthalmologic assessment, significant proptosis of the right eye was noted. Ocular motility on the affected side was severely restricted. The right eye was non-responsive to light stimuli, indicating blindness with no light perception and absence of pupillary reaction. Furthermore, severe dental caries and toenail onychomycosis were observed. Neither meningeal signs nor focal neurological deficits were present. His BMI was 20.1 kg/m^2^.

Endoscopic evaluation revealed a pronounced left-sided deviation of the nasal septum, consistent with a severe post-traumatic deformity. The right nasal cavity was completely filled with pathological tissue displaying fungal morphology. The surrounding mucosa appeared markedly inflamed and edematous, suggesting advanced fungal-inflammatory involvement.

Laboratory tests revealed mild anemia (Hb 12.4 g/dl, normal MCV), hyperglycemia of 18.2 mmol/l, glucosuria, glycated hemoglobin (HbA1c) of 16% and normal renal function. Aspergillus antigen was negative (Aspergillus galactomannan antigen detecting assay was performed using PlateliaTM Aspergillus Ag test, Bio-Rad, Marnes-la-Coquette, France).

Initial, pre-operative CT scans with massive inflammatory opacifications are presented in Figure 2.

### 2.2. Operative Report: Urgent Endoscopic Sinus Surgery

The patient was scheduled for urgent endoscopic debridement due to suspected fungal sinonasal infection. Following decongestion (anemization) of the nasal mucosa on the right side, extensive inflammatory lesions consistent with fungal morphology were visualized. These changes had caused significant destruction of the middle nasal concha as well as the ethmoidal labyrinth. Endoscopic removal of the pathological tissue was performed, encompassing both the anterior and posterior ethmoid sinuses. The natural openings of the maxillary, sphenoid, and frontal sinuses were subsequently identified, surgically opened, and widened. Fungal lesions were thoroughly debrided from the maxillary sinus. Upon accessing the frontal sinus, purulent discharge was encountered and evacuated.

During further inspection, a bony defect of the lamina papyracea was identified in its anterior portion, measuring approximately 4 mm in diameter. The lamina papyracea was resected further, extending posteriorly to the level of the posterior ethmoid cells. A horizontal incision was then made in the orbital periosteum, which allowed for partial herniation of the orbital fat into the ethmoid cavity—facilitating access and drainage. Anterior nasal packing using Merocel was placed to support mucosal healing and maintain patency. Figure 3, Figure 4 and Figure 5 present the intraoperative view.

Histopathological examination of the specimens from ethmoid and maxillary sinuses confirmed fungal infiltration, described as “mixed-cell infiltration with fragments of hyphae (most likely belonging to the genus *Mucor*)”. Direct KOH microscopy was performed and revealed fungal hyphae. Tissue fragments collected during the first surgery were inoculated on Sabouraud glucose agar with gentamicin and chloramphenicol (bioMerieux, Marcy-l’Etoile, France) and on Sabouraud dextrose broth (Biomaxima, Lublin, Poland). Cultures were incubated at 25 °C and 35 °C. The significant growth of *Rhizopus arrhizus* was obtained from all samples. Colonies were very fast growing, white cottony at first, becoming brownish-gray to blackish-gray (Figure 6). Non-septate sporangiophores ending with spore-filled ellipsoidal sporangia as well as non-septate hyphae and rhizoids were observed in slides prepared with Lactophenol blue solution (Sigma-Aldrich, manufactured by Merck, Darmstadt, Germany)—presented in Figure 7. Species’ identification was also confirmed using matrix-associated laser desorption ionization time-of-flight mass spectrometry (MALDI Biotyper^®^ Sirius, Bruker Daltonics, Brema, Germany). The antifungal susceptibility testing was performed using RPMI agar plates (bioMerieux, Marcy-l’Etoile, France) and MIC Test Strips (Liofilchem, Roseto degli Abruzzi, Italy) and the results were the following: amphotericin B MIC = 0.75; itraconazole MIC = 4; posaconazole MIC = 0.75; and isavuconazole MIC = 2. While molecular methods are considered the reference standard for fungal species’ identification, they were not available at our center. The combination of the aforementioned assessments, however, enabled reliable identification of the pathogen.

### 2.3. Medical Treatment

After the surgery, the patient was transferred to non-surgical wards (Internal Medicine, Metabolic Diseases and Diabetology Department, followed by the Infectious Diseases Department), where the multidisciplinary care was continued. Viral hepatitis and HIV serology were negative. To evaluate for humoral immune deficiency, complement components and immunoglobulin levels were measured and found to be within normal limits.

Computed tomography (CT) of the abdomen revealed chronic pancreatitis, right renal agenesis and aortic atherosclerosis. Echocardiography showed no abnormalities and an ejection fraction of 65%. There were no neurological or nephrological complications related to diabetes mellitus. There were no clinical features of chronic liver disease or cirrhosis.

Magnetic resonance imaging of the head revealed diffuse bilateral cortico-subcortical malacia with surrounding gliosis, suggestive of secondary hemorrhagic transformation of small ischemic foci.

Orbital MRI demonstrated a 23 × 15 × 14 mm abscess in the upper-medial quadrant of the right orbit, showing strong contrast enhancement and centrally elevated T2 signal. The lesion was adjacent to an area of infiltration, which also exhibited contrast enhancement and restricted diffusion. The process extended into the orbital adipose tissue, extraocular muscles and nasal cavity, causing osseous destruction of ethmoid cells, superior and middle nasal conchae, as well as the wall of the right maxillary sinus. The suspicion of erosion of the medial orbital wall (of the right orbit) was raised and later confirmed in CT imaging. Posteriorly, the infiltration progressed into the optic canal, involving the optic nerve, which appeared thickened and showed contrast enhancement. Additionally, a massive right maxillary sinus opacification and mucosal thickening were present.

The treatment was initiated with intravenous (IV) liposomal amphotericin B at a dose of 5.5 mg/kg of body weight once daily, oral posaconazole (400 mg twice daily), and intensive insulin therapy with multiple daily injections of a rapid-acting insulin analog and one basal injection of long-acting insulin.

The therapy was complicated by renal insufficiency. Initial eGFR (glomerular filtration rate by CKD-EPI equation) was 125 mL/min/1.73 m^2^, but the gradual deterioration of creatinine clearance was observed from week 2. Despite prophylactic volume expansion with isotonic fluids prior to amphotericin B administration, after 4 weeks of treatment, eGFR declined below 50 and reached 38 mL/min/1.73 m^2^, after 3 more weeks. At this point, a decision was made to discontinue amphotericin B and posaconazole, and to switch to intravenous (IV) isavuconazole. The initial loading dose was 200 mg every 8 h for six doses, followed by a maintenance dose of 200 mg once daily. The antifungal dosage was consulted with a clinical pharmacologist and remained unchanged throughout the treatment. At our center, therapeutic drug monitoring of these agents was not accessible.

During the sixth week of hospitalization, the patient developed upper abdominal pain, nausea and retrosternal burning of acute onset. Gastroscopy revealed esophageal ulcerations and macroscopic features of gastropathy. Histopathological analysis confirmed low-grade hemorrhagic gastritis and herpetic esophagitis. The treatment was supplemented with a 21-day course of IV acyclovir and a few weeks of proton pomp inhibitor, leading to the complete resolution of symptoms.

In week 10 of the therapy, a follow-up MRI of the head demonstrated no conspicuous changes. In the MRI of the orbits, an abscess appeared stable in size, with a slightly smaller area of central T2 signal elevation. The infiltration extending toward the optic canal, associated with bony destruction, continued to exhibit contrast enhancement and unchanged measurements.

### 2.4. Otolaryngological Consultations and Follow-Up

The patient underwent multiple otolaryngological consultations, during which repeated endoscopic assessments of the local condition were performed. Edematous changes in the mucosal lining within the right maxillary sinus and ethmoid complex were observed. The anatomical structures appeared significantly altered due to an aggressive inflammatory process and the subsequent surgical intervention.

During follow-up visits, the nasal cavities were regularly debrided and cleared of retained secretions to improve ventilation and reduce the inflammatory burden.

Due to retinal artery occlusion and vision loss in the right eye, ophthalmology specialists were involved in the patient’s multidisciplinary care. However, visual recovery was deemed very unlikely, and the patient showed no improvement throughout the treatment.

Given the suboptimal response to antimicrobial therapy after more than 12 weeks, the patient was considered a candidate for surgery.

### 2.5. Second Surgical Procedure

During the second endoscopic procedure, significant postoperative anatomical alterations were observed, including extensive scarring and multiple synechiae. A defect of the medial orbital wall was identified. The underlying bone was visualized and found to be covered only by periosteum. Using a diamond burr, a portion of the bony boundary of the frontal recess was carefully removed to create a surgical access route toward the lesion previously identified on CT and MRI imaging. The lesion was located in the superomedial quadrant of the right orbit, between the medial and superior rectus muscles and adjacent to the retrobulbar segment of the optic nerve. The orbital periosteum was incised, and orbital fat was gently displaced to access the lesion. The pathological tissue was successfully removed and sent for histopathological examination. A nasal dressing was applied to complete the procedure.

After the surgery, the medical therapy with isavuconazole was continued. The patient was discharged 95 days after the initial presentation and continued treatment with posaconazole oral formulation (400 mg twice daily). Careful surveillance of the patient’s glycemia and insulin dosing during the hospitalization led to significant improvement in diabetes control parameters, with HbA1c decreasing gradually to 10.3% in week 8, then 7.5% in week 12 and eventually to 6.7% in week 17. The patient was educated on the importance of proper insulin therapy.

On follow-up three weeks and then two, and four months later he presented in a good general condition, with no new symptoms. Three follow-up MRI scans (after 16, 25 and 33 weeks of therapy) showed a further approximately 30% decrease in abscess size and reduced muscle infiltration. Optic nerve involvement was still visible, as were sinus opacification and mucosal thickening.

The histopathological report from the second surgery revealed mixed-cell infiltration and fibrosis (interpreted as features of chronic sinusitis). In one of the three specimens, supplementary GMS (Grocott–Gömöri’s methenamine silver stain) and PAS (Periodic acid–Schiff) staining demonstrated single fungal structures morphologically consistent with fungi of the class Mucormycetes. Microbiological culture obtained during the second surgery was negative.

### 2.6. Postoperative Endoscopic Follow-Up

Postoperative endoscopic evaluations demonstrated a normal healing process, with no evidence of pathological secretions and maintained patency of the sinus outflow tracts (Figure 8). However, the mucosal lining of the maxillary and ethmoid sinuses remained thickened, consistent with ongoing post-inflammatory changes.

Fluticasone nasal spray and nasal irrigation were recommended together with continuation of oral posaconazole.

Figure 9 represents a graphic summary of the patient’s treatment.

## 3. Discussion

We provide a detailed description of the clinical course of rhino-orbital-cerebral mucormycosis together with an in-depth analysis of both surgical and medical aspects of its management.

Our patient, a 45-year-old man with uncontrolled diabetes, fits the most typical demographic and epidemiological profile of the infection as described in a recent global meta-analysis [11]. However, the prevalence of diabetes and other underlying conditions (hematological malignancy, solid organ transplantation, trauma, chronic kidney disease, alcohol consumption) is heavily dependent on geographic factors [3,12].

Rhino-orbital-cerebral mucormycosis usually manifests rapidly with fever, headache, nasal congestion and facial pain, which can later be accompanied by visible tissue necrosis, characteristic black eschar and cranial nerve involvement [12,13]. With the exception of an external necrotic lesion, our patient presented with the most common symptoms of the disease, which may have contributed to timely diagnosis and prompt management.

Due to its relatively low prevalence, mucormycosis is sometimes overlooked in the differential diagnosis until a pathological report suggests a fungal etiology. Diagnosis may then rely on histopathology alone; however, cultures should always be pursued to identify the pathogen and tailor antifungal therapy. Serological tests, like galactomannan and beta-D-glucan are negative [11,13]. *Rhizopus arrhizus* is the most frequently isolated causative agent of mucormycosis and genus *Rhizopus* in general—most commonly associated with rhino-orbital-cerebral form of the disease [14].

Untreated mucormycosis is almost universally fatal [15]. The mainstays of treatment include antifungal therapy, surgical debridement, and management of immunosuppression. Liposomal amphotericin B is the drug of choice, with a recommended dose of 5–10 mg/kg administered throughout the treatment course. Stepwise escalation of the dose as well as amphotericin deoxycholate should be avoided. In cases involving the central nervous system, the full dose of 10 mg/kg is recommended [4]. Posaconazole and isavuconazole are new triazoles clinically active against Mucorales. They play a role in mucormycosis management as alternatives when first-line therapy fails, in cases of amphotericin B–associated toxicity (particularly renal), and as a step-down treatment once a clinical response is achieved [4,13]. Initiating therapy with a combination of amphotericin B and oral posaconazole has been described and shown to be beneficial in small studies, although it has not been officially endorsed by current guidelines. This approach is not expected to increase toxicity and may be considered on a case-by-case basis [4,13]. Echinocandins and other azoles have no proven activity against Mucorales [13]. The optimal timing for switching from intravenous amphotericin B to oral maintenance therapy, as well as the total duration of treatment, have not been clearly established. These treatment parameters vary widely—from several weeks to as long as three years—depending on the form of the disease, the degree of immunosuppression, and whether surgical intervention was performed. Decisions regarding treatment duration should be guided by both clinical and radiological response [4,13].

In our patient, the treatment was started with a combination of amphotericin B and oral posaconazole, and continued for 58 days, then switched to intravenous isavuconazole due to renal insufficiency. Amphotericin B therapy is associated with a significant risk of nephrotoxicity, typically presenting as elevated serum creatinine levels. Electrolyte disturbances, such as hypokalemia and hypomagnesemia, may also be present. Preventive measures include appropriate fluid administration prior to drug infusion [16]. In most cases, renal impairment is reversible upon discontinuation of amphotericin B. Nephrotoxicity is certainly less frequently observed in patients treated with liposomal formulation of the drug [17].

After hospital discharge, the patient continued therapy with posaconazole oral suspension, which is the only drug available for use outside the hospital setting. The oral suspension of posaconazole has suboptimal bioavailability and is inferior to the delayed-release tablet formulation, which, unfortunately, is not available in Poland [4,13]. Treatment with posaconazole oral suspension (outside the hospital) is almost entirely covered by the public payer only for patients with hematological malignancy. No reimbursement is provided for individuals with other risk factors who develop mucormycosis. The monthly cost of treatment for our patient is approximately EUR 900–1000.

Isavuconazole (brand name Cresemba) has been proven to have similar efficacy and safety to amphotericin B [18]. Oral isavuconazole has a bioavailability comparable to intravenous formulation [19]. It has been approved by the European Medicines Agency for the treatment of mucormycosis and invasive aspergillosis, and has been granted an ‘orphan drug’ status. In Poland, isavoconazole is available only for hospital use and has been incorporated into our patient’s treatment after he developed renal insufficiency.

Surgical management plays a crucial role in mucormycosis therapy. It not only complements medical treatment, but markedly alters the prognosis in the rhino-orbital-cerebral form of the disease. In a recent cohort study, all patients who underwent immediate orbital exenteration survived the 3-year follow-up period despite extensive disease [20]. According to the literature, aggressive debridement of all infected tissues may reduce the mortality by nearly 50% [21].

Endoscopic surgery is a very important and effective component—alongside systemic treatment—in the management of the rhino-orbital form of mucormycosis. It is a safe method that allows for precise visualization and thorough removal of inflammatory and necrotic foci. The procedure is minimally invasive and causes limited tissue trauma. Compared to open surgery, it has been linked to lower morbidity and recurrence rates [22].

Our patient is a forty-five-year-old man with diabetes mellitus secondary to chronic pancreatitis. He developed a rare infectious complication frequently attributable to poor glycemic control. The patient claimed that the diabetes had been diagnosed one year prior to admission, but he had not started treatment or maintained a diet because of his work requiring frequent travels abroad. The additional challenge in this case might have been the fact that the patient was not eligible for oral treatment and struggled to incorporate insulin therapy into his daily life. This case should draw attention to the significance of patient education and adherence in the therapy of diabetes mellitus. Due to wide fluctuations in the patient’s glucose levels observed during the hospitalization (at least partly related to an inconsistent diet), the patient could probably benefit from an insulin pump. However, there are specific considerations related to insulin pump therapy. In Poland, it is reimbursed by the public payer—the National Health Fund (NFZ)—only for patients below the age of 26 and its cost can be a limiting factor (approximately EUR 3000 + maintenance). Furthermore, our patient’s poor understanding of the disease and the principles of insulin dosing (including episodes of hypoglycemia), as well as a history of alcohol misuse, would make insulin pump use questionable and require additional training and support. The abovementioned circumstances may impair the effectiveness and safety of insulin pump therapy and, according to the Polish Diabetes Society, serve as contraindications in the context of potential reimbursement [23].

The only oral medication that could facilitate the patient’s diabetes control was SGLT2 inhibitor, which in Poland also has a restricted reimbursement policy and would need to be covered by the patient.

## 4. Conclusions

We presented the case of a patient with uncontrolled diabetes who was diagnosed with mucormycosis—a rare but emerging fungal infection. Its treatment is prolonged, costly, and complex, involving both antifungal pharmacotherapy and often extensive surgical procedures. Successful outcomes rely on prompt diagnosis and multidisciplinary approach, with close collaboration among infectious disease specialists, surgeons, and other relevant clinicians. By presenting this case, we aim to raise awareness of invasive fungal infections, which are likely to become more prevalent with the rising burden of immunosuppression, malignancy and diabetes.

## Figures and Tables

**Figure 1 microorganisms-13-02695-f001:**
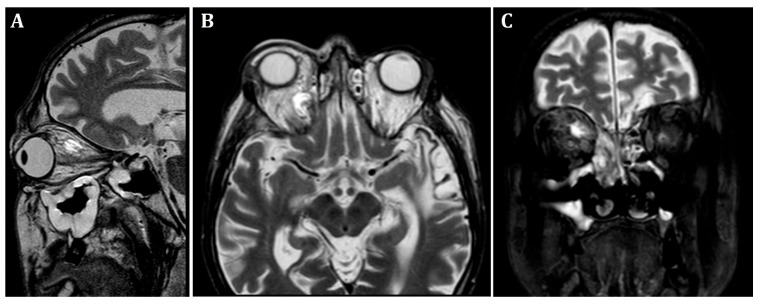
Initial pre-operative magnetic resonance imaging. T2-weighted MRI scans of the orbits. (**A**)—sagittal, (**B**)—frontal, (**C**)—transverse planes. Visible infiltration of extraocular muscles and optic nerve. Inflammatory opacifications in ethmoid cells and maxillary sinus, extending into the orbital adipose tissue and extraocular muscles, causing osseous destruction of ethmoid cells.

**Figure 2 microorganisms-13-02695-f002:**
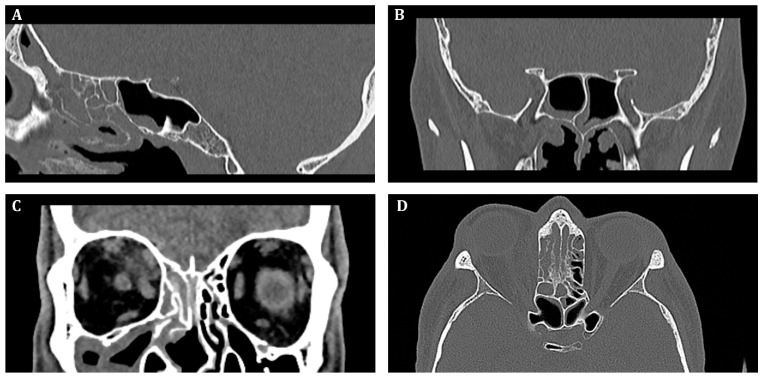
Pre-operative computed tomography of the sinuses, (**A**)—sagittal, (**B**,**C**)—frontal, (**D**)—transverse scan. Inflammatory opacification in frontal, maxillary and ethmoid sinus on the right side. Suspected abscess in the upper-medial quadrant of the right orbit.

**Figure 3 microorganisms-13-02695-f003:**
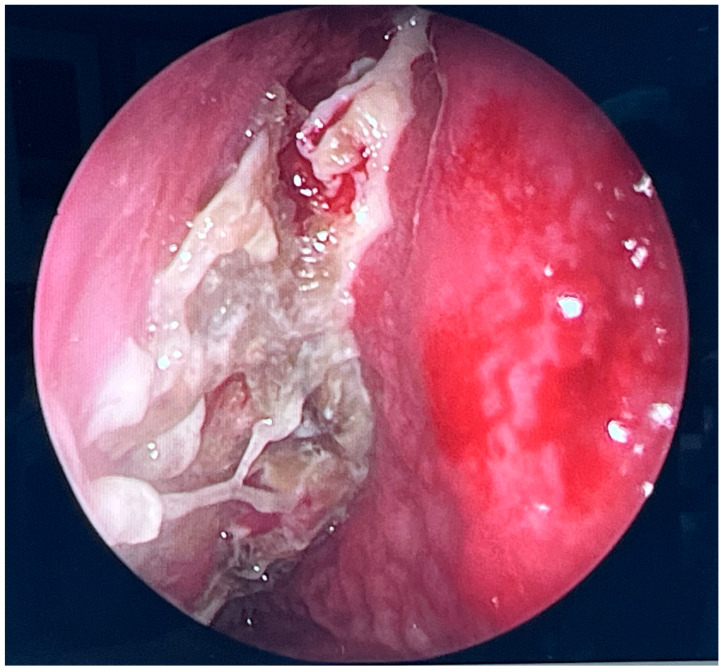
Endoscopic surgery: intraoperative view showing nasal cavity filled with fungal masses.

**Figure 4 microorganisms-13-02695-f004:**
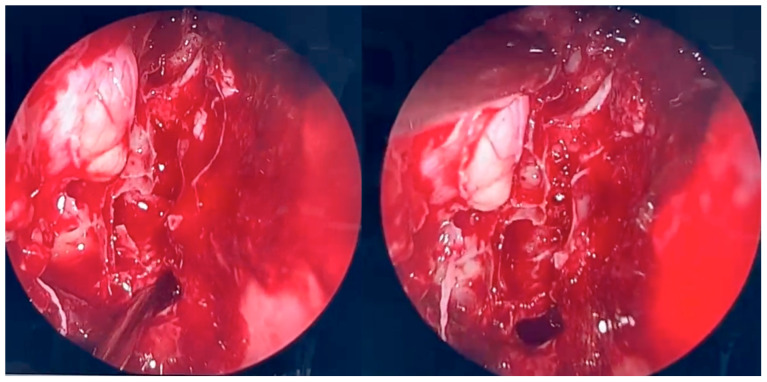
Endoscopic surgery: intraoperative view after removal of fungal infiltration to the level of the skull base. Opening of the sphenoidal sinus seen inferiorly (6 o’clock position). The upper-left part of the picture (11 o’clock position) shows the site of orbital periosteum incision with partial herniation of orbital fat into the ethmoid cavity. Two images obtained from the same projection.

**Figure 5 microorganisms-13-02695-f005:**
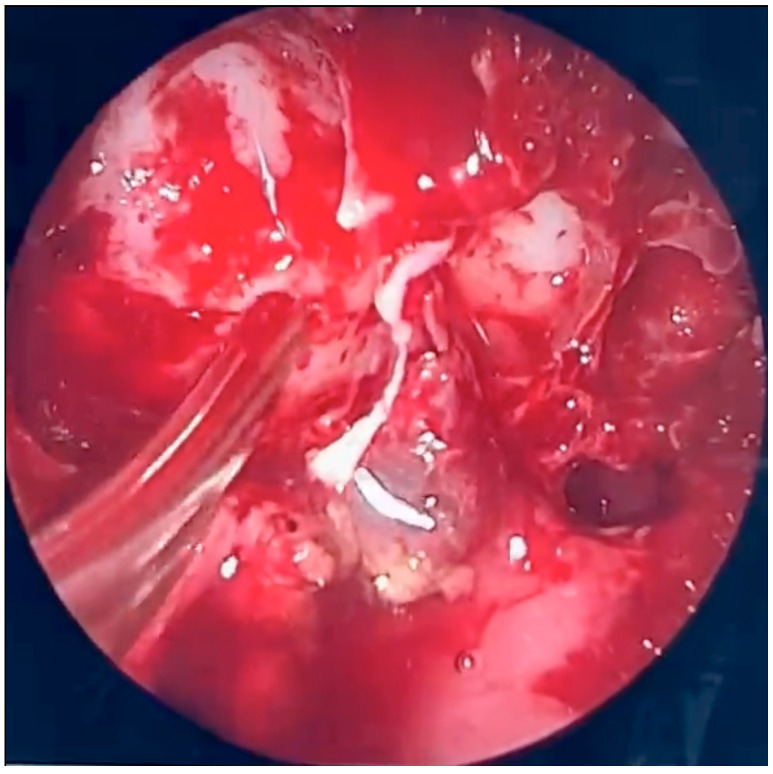
Endoscopic surgery: intraoperative view of the right maxillary sinus after removal of fungal masses. Opening of the sphenoidal sinus visible on the right (4 o’clock position).

**Figure 6 microorganisms-13-02695-f006:**
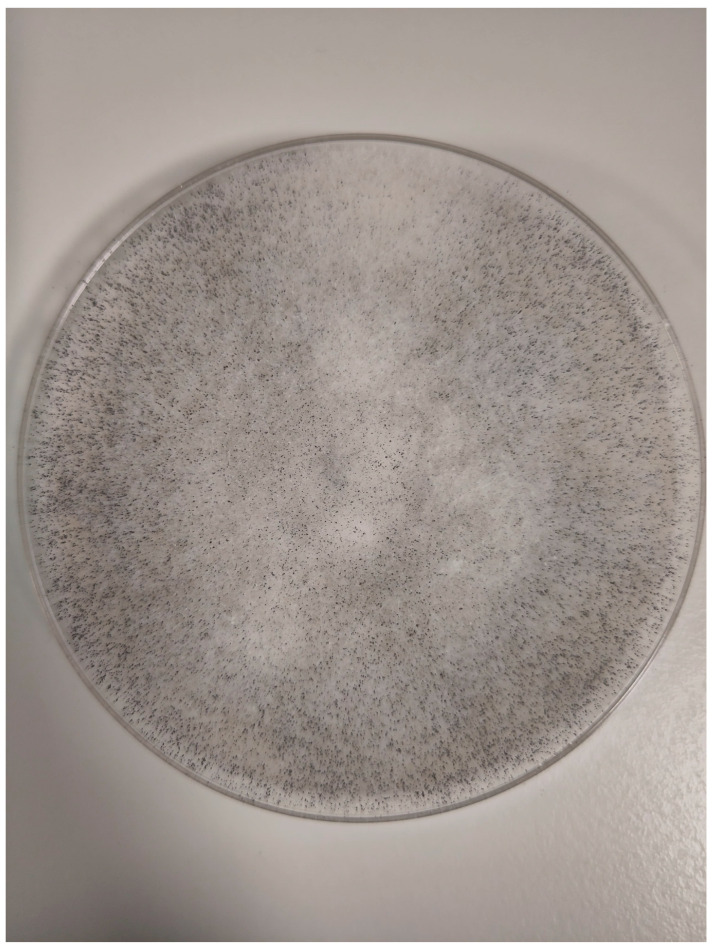
*Rhizopus arrhizus* colonies.

**Figure 7 microorganisms-13-02695-f007:**
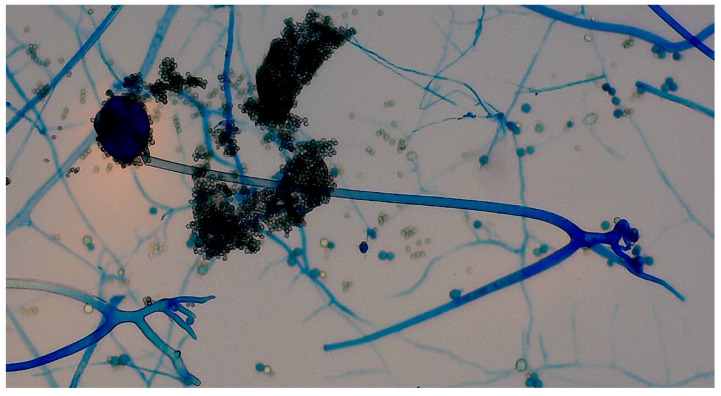
Slides prepared with Lactophenol blue solution showed non-septate sporangiophores ending with spore-filled ellipsoidal sporangia as well as non-septate hyphae and rhizoids.

**Figure 8 microorganisms-13-02695-f008:**
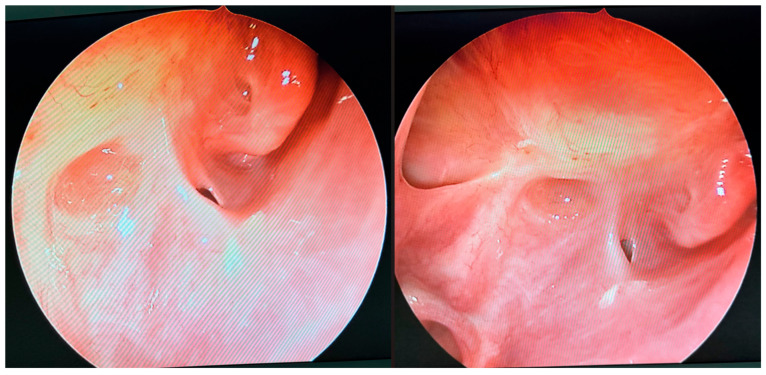
Endoscopic follow-up view 33 weeks after initial surgery. The openings of the maxillary sinus, orbit, and sphenoidal sinus are conspicuous. No inflammatory lesions or pathological discharge are present. The local condition appears unremarkable.

**Figure 9 microorganisms-13-02695-f009:**
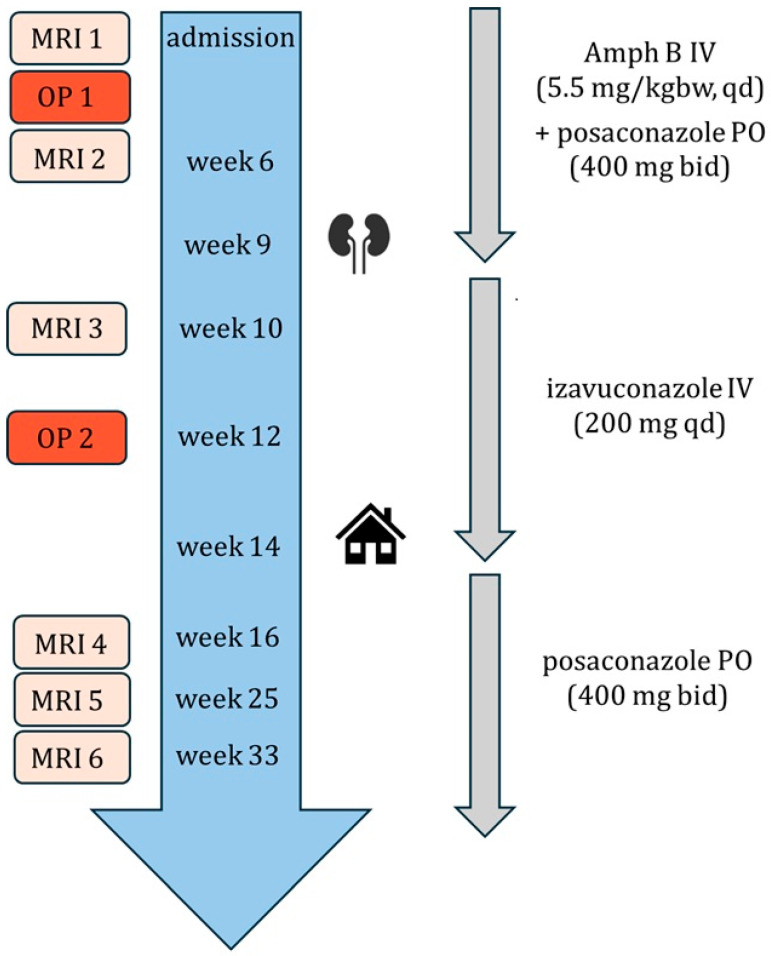
Graphic summary of the patient’s treatment. MRI = magnetic resonance imaging. OP = surgery. Amph B = amphotericin B. PO = oral. IV = intravenous. kgbw = kilogram of body weight. qd = once daily. bid = twice a day. Kidney symbol = renal insufficiency. Home = discharge.

## Data Availability

The original contributions presented in this study are included in the article. Further inquiries can be directed to the corresponding author.

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
