# Peer review of "Rhino-Orbital-Cerebral Mucormycosis Complicated by Vision Loss in a Patient with Uncontrolled Diabetes: A Case Report"

_microorganisms, 2025, doi:10.3390/microorganisms13122695_

Round 1
Reviewer 1 Report
Comments and Suggestions for Authors
The case describes a severe and life-threatening infection, which is interesting. However, The report suffers from a critical lack of novelty. As a journal of microorganisms, the case lack details description of the fungus.
- The report states that histopathology "confirmed fungal infiltration" after the first surgery but provides no critical details. No standard fungal stains (e.g., GMS, PAS) in the initial assessment, nor a description of the hyphal morphology (e.g., broad, aseptate, ribbon-like hyphae with right-angle branching) essential for differentiating mucormycosis from other molds. The later mention of positive GMS/PAS stains is from an ambiguous finding ("single fungal structures") and is not adequately interpreted.
- No Direct Microscopy: There is no description of a direct microscopic examination (e.g., KOH mount, Calcofluor white) of the clinical specimens. This is a fundamental, rapid diagnostic tool for presumptive diagnosis and its absence represents a significant gap in the diagnostic timeline and reporting.
- No slide culture for morphological identification and no photo of colony.
- Lack of Gold-Standard Molecular Identification: The identification of Rhizopus arrhizus relies on MALDI-TOF MS without confirmation by ITS sequencing, which is the molecular gold standard for definitive species-level identification of mucorales.
Author Response
The Authors would like to express their sincere gratitude to the Reviewer for their time and expertise. We appreciate the careful review and believe that our manuscript has been improved. We respond to each comment below and enclose the revised version of the manuscript.
- We thank the Reviewer for pointing out inconsistencies in histopathological descriptions. The initial histopathological report was relatively concise and stated: “mixed-cell infiltration with fragments of hyphae (most likely belonging to the genus Mucor). The description of fungal stains and hyphal morphology was not provided. We expanded the description in lines 154-156 accordingly. We also enclosed a more detailed description of the second histopathological report with special emphasis on GMS/PAS staining (lines 261-265).
- We agree this section needed a more elaborate explanation. Direct KOH microscopy was performed and revealed fungal hyphae. In our laboratory, direct microscopy is performed whenever possible to aid the final interpretation of the examination. It is not, however, routinely included as part of the microbiological report. We provide additional explanation in lines 156-157.
- We appreciate the Reviewer’s insightful comment. Slide culture is frequently employed in our center as a confirmatory tool when other examinations yield inconclusive or contradictory results; however, this was not the case for the present patient. The culture enabled precise identification of Rhizopus arrhizus based on the colony’s morphological characteristics and the microscopic features observed in slides prepared with Lactophenol blue solution. Species identification was additionally confirmed using matrix-assisted laser desorption/ionization time-of-flight mass spectrometry (MALDI Biotyper). We believe that refraining from performing a slide culture was justified in this instance, as it would have unnecessarily prolonged the identification process while probably not altering the result.
Unfortunately, we do not have a photograph of the colony readily available. However, we could obtain one from a banked sample within a few days, should the Reviewer and Editor consider that it would enhance the manuscript. - We appreciate this comment. While molecular methods are considered the reference standard for fungal species identification, they are not available in our center. The combination of morphological assessment and MALDI-TOF MS analysis, however, enabled a reliable identification in this case. We have modified the corresponding section to better explain this issue in lines 170-173.
Reviewer 2 Report
Comments and Suggestions for Authors
This manuscript presents a clinical case of mucormycosis in a diabetic patient. The authors provide an extensive description of the clinical, radiological as well as therapeutic characteristics of this patient. The use of antifungal drugs and surgical interventions is described in detail. They mention the difficulties they had in controlling his blood glucose level and the economic aspects they had to bear. The diagnosis of mucormycosis was based on accepted mycologic studies and histopathology. As a case report, it does not include original findings, but it is a very good presentation. I do not consider myself capable of judging the quality of their English.
Author Response
The Authors would like to sincerely thank the Reviewer for their time and expertise. We are grateful for the positive evaluation of our work.
Reviewer 3 Report
Comments and Suggestions for Authors
Lara and colleagues submit a case report regarding mucormycosis complicated by vision loss in a patient with uncontrolled diabetes.
Short summary of the manuscript explaining what the study is about: This is a case report regarding rhino-orbital-cerebral mucormycosis complicated by vision loss in a patient with uncontrolled diabetes. The authors are in Poland.
Comments:
Methodology: This is a narrative format case report.
Figure interpretation: excellent figures. In Figure 7, add dosages for the three antifungal agents. Also place arrows for the times and results of any therapeutic drug monitoring.
Minor comments:
- Rhizopus arrhizus should be in italics, as they are Latin genus and species names. This occurs on line 158, 290, 291, etc.
- In section 2.2, “amfotericin” should be spelled amphotericin
- On line 178, “clinical features or chronic liver disease” should be “clinical features of chronic liver disease”.
- On line 193, please add dosages for liposomal amphotericin B and posaconazole. Were any blood levels of the antifungal medications checked, and were they either subtherapeutic or supratherapeutic? Did changes in medication dosages ensue based on the blood levels?
- On line 202, please add the dosage of Isavuconazole. Were any blood levels of the antifungal medications checked, and were they either subtherapeutic or supratherapeutic? Did changes in medication dosages ensue based on the blood levels?
- If no therapeutic drug monitoring was performed, report why this was the case.
- Please list any serial results of HgbA1c levels, showing that the diabetes was coming under control over the course of time.
Author Response
The Authors sincerely thank the Reviewer for their time and positive feedback. We appreciate the careful evaluation and believe that our manuscript has been improved. We enclose the revised manuscript.
We are thankful for spotting the spelling errors. We have corrected Rhizopus and Rhizopus arrhizus in lines 160, 304, 305 as well as italicized genera names in line 40. We have also corrected the misspelling of “amphotericin” in line 169 and ensured it does not appear elsewhere in the manuscript. We have corrected the typo in line 183.
Thank You for pointing out the issue of the therapeutic drug monitoring. Unfortunately, it was not available in our hospital. We added an explanation in lines 209-211. We also added information on dosage of antifungals in lines 199, 208-209, and 251, and in the Figure 7.
We appreciate the Reviewer's insight regarding HgbA1c levels. We have now included the relevant information in lines 253-254.